# From Bench to Bedside: Translational Approaches to Cardiotoxicity in Breast Cancer, Lung Cancer, and Lymphoma Therapies

**DOI:** 10.3390/cancers17071059

**Published:** 2025-03-21

**Authors:** Valerio Nardone, Dafne Ruggiero, Maria Giovanna Chini, Ines Bruno, Gianluigi Lauro, Stefania Terracciano, Angela Nebbioso, Giuseppe Bifulco, Salvatore Cappabianca, Alfonso Reginelli

**Affiliations:** 1Department of Precision Medicine, University of Campania “L. Vanvitelli”, 80138 Naples, Italy; valerio.nardone@unicampania.it (V.N.); druggiero@unisa.it (D.R.); angela.nebbioso@unicampania.it (A.N.); salvatore.cappabianca@unicampania.it (S.C.); alfonso.reginelli@unicampania.it (A.R.); 2Department of Pharmacy, University of Salerno, Via Giovanni Paolo II, 132, 84084 Fisciano, Italy; brunoin@unisa.it (I.B.); glauro@unisa.it (G.L.); sterracciano@unisa.it (S.T.); bifulco@unisa.it (G.B.); 3Department of Biosciences and Territory, University of Molise, Contrada Fonte Lappone, Pesche, 86090 Isernia, Italy

**Keywords:** cardiotoxicity, preclinical models, chemotherapy, radiotherapy, immunotherapy

## Abstract

Cancer treatments such as chemotherapy, radiotherapy, and immunotherapy can sometimes harm the heart, a condition known as cardiotoxicity, which affects patients’ health and complicates treatment strategies. This review explores how laboratory (in vitro) and animal (in vivo) models help scientists understand the causes of cardiotoxicity and develop ways to prevent or reduce its impact. In vitro studies focus on how cancer drugs or radiation affect heart cells, revealing important details about cellular damage. In vivo models allow researchers to study how these treatments affect the whole heart over time, offering insights into chronic heart problems caused by cancer therapies. By combining data from these models, researchers can uncover the mechanisms of cardiotoxicity and design effective prevention strategies, ultimately improving patient outcomes and advancing cancer care.

## 1. Introduction

Thoracic tumors, including lung cancer, breast cancer, and childhood lymphomas, exhibit distinct epidemiological patterns and survival outcomes. Beyond age, sex, and gender, additional factors such as comorbidities and concomitant medications, including prescribed drugs, can significantly influence both the efficacy and the side effects of oncologic treatments, highlighting the need for a comprehensive patient assessment [1,2,3]. Lung cancer remains one of the leading causes of cancer-related mortality worldwide, with a 5-year survival rate of approximately 65% for localized tumors. However, this rate drops significantly in advanced stages, particularly for cases with distant metastases, where survival rates range from 8% to 25% [4,5,6]. In contrast, breast cancer patients, especially those with localized disease, have higher survival rates, with 5-year survival rates exceeding 90% in many regions [7,8,9]. Childhood lymphomas, particularly Hodgkin lymphoma, also show favorable long-term survival outcomes, with 5-year survival rates approaching 85% or higher [10,11,12]. While these tumors show relatively high survival rates, survivors, particularly those with breast cancer and childhood lymphoma, face an increased risk of chronic cardiovascular complications due to the long-term effects of treatments such as chemotherapy and radiation. These side effects can be asymptomatic, such as a reduction in Left Ventricular Ejection Fraction (LVEF), or they can be symptomatic, including tachycardia (rapid heartbeat), arrhythmia (irregular heartbeat), and cardiomyopathy (weakening of the heart muscle) (Figure 1) [13]. The risk and severity of cardiotoxicity can depend on various factors, including the specific type of cancer treatment, the duration and dosage of treatment, the patient’s age, and pre-existing heart conditions. Close monitoring of cardiac function is often necessary during and after cancer treatment to detect any signs of cardiotoxicity and manage any related symptoms [14].

Preclinical studies employing animal models and pharmacologic interventions were selected based on their ability to replicate key pathological mechanisms of cardiotoxicity observed in clinical settings. Particular emphasis was placed on models closely mimicking human cardiac physiology and disease progression, allowing for a more reliable translation of findings to clinical practice. Additionally, we prioritized studies that evaluated pharmacologic strategies with potential clinical applicability, including both cardioprotective agents and targeted interventions aimed at mitigating treatment-induced cardiac dysfunction.

Concerning radiotherapy, Radiation-Induced Heart Disease (RIHD), that is to say, all cardiac complications related to Radiation Therapy (RT), represents one of the most serious causes of morbidity and mortality among cancer survivors. The development of RIHD depends on several determinants, such as the total radiation dose, the dose per fraction, the volume of the heart irradiated, the type of radiation used, and individual patient characteristics. Advances in radiation technology, such as intensity-modulated radiation therapy (IMRT) and proton therapy, have allowed for more precise targeting of tumors while sparing healthy tissues, including the heart. These techniques have contributed to a reduction in the incidence of RIHD. However, despite these advancements, RIHD remains a concern [15]. Cancer survivors who have undergone radiation therapy, particularly those treated for breast cancer, lymphomas, or lung tumors, are at a higher risk of developing heart-related complications later in life. The effects of radiation on the heart can take several years or even decades to become clinically evident. For all these reasons, further efforts to refine radiation techniques, the improvement of cardioprotective strategies, and long-term monitoring are essential to decrease the incidence and impact of RIHD in the future [16].

Antineoplastic drugs, including doxorubicin, can have significant side effects, including acute and chronic cardiotoxicity. Doxorubicin is known to cause acute cardiotoxicity within 2–3 days after administration in approximately 11% of patients, while chronic cardiotoxicity occurs in about 1.7% of patients [17,18]. Newer anticancer drugs, such as VEGF inhibitors, can also increase the risk of myocardial ischemia and have been associated with hypertension and heart failure. VEGF inhibitors work by blocking the growth of new blood vessels, which can lead to reduced blood flow to the heart and other organs, which can cause chest pain, shortness of breath, and other symptoms. In addition, VEGF inhibitors have been associated with an increased risk of hypertension, which can further increase the risk of heart problems. Heart failure, a condition in which the heart cannot pump blood effectively, has also been reported in patients receiving VEGF inhibitors [19,20].

Moreover, the administration of immune checkpoint inhibitors (ICIs) can lead to adverse cardiovascular effects such as myocarditis, pericarditis, and cardiomyopathy. Some of the known risk factors for ICIs-induced cardiotoxicity include pre-existing cardiac disease, older age, high baseline troponin levels, and prior treatment with anthracyclines or radiation therapy to the chest. Other factors that may contribute to the development of cardiotoxicity with ICIs include the type of cancer being treated, the dose and duration of treatment, and the specific type of ICI being used [21,22,23].

To understand the potential cardiotoxic effects of various drugs or interventions, a wide range of multidisciplinary studies have been conducted in these years using both in vitro models (cell cultures) and in vivo models (animal models). These models have provided valuable insights into the mechanisms of cardiotoxicity and have helped researchers develop focused strategies to mitigate or prevent such effects. In vitro models involve culturing cardiac cells, such as cardiomyocytes, in a laboratory setting. By exposing the cells to different concentrations of drugs or other substances, researchers can observe changes in cell viability, function, and other relevant parameters. In vitro models also enable the investigation of specific cellular mechanisms underlying cardiotoxicity [24]. Animal models, on the other hand, provide a more complex and holistic view of cardiotoxicity. Through in vivo studies, the evaluation of systemic effects of potential cardiotoxic substances or interventions can be detected. These models can provide insights into the overall cardiovascular function, including heart rate, blood pressure, electrocardiogram (ECG) changes, and other relevant physiological parameters. They also enable the assessment of long-term effects, such as chronic toxicity or the development of cardiac diseases. By combining data from both in vitro and in vivo studies, researchers can gain a comprehensive understanding of cardiotoxicity, identify potential mechanisms of damage, and develop strategies to prevent or minimize adverse effects (Figure 2) [25,26,27,28,29].

## 2. In Vitro Models

In vitro models have been developed to study the effects of anticancer therapies on various cellular and tissue properties. These models can be used to investigate the mechanisms underlying the observed toxicities, as well as to test potential therapeutic interventions. Cell line assays are widely used in biomedical research to evaluate the effects of drugs on cellular processes, such as contractility and electrophysiological properties, as well as pharmacokinetics, cytotoxicity, and the efficacy of antineoplastic treatments. Due to the presence of new cancer therapies based on both traditional drugs and radiotherapy, there is a growing need for robust preclinical screening approaches for the analysis of cardiotoxicity-associated cancer treatments prior to human clinical trials [30]. Recently, the use of hiPSC-CMs (human induced pluripotent stem cell-derived cardiomyocytes) and human heart slices, an advantageous technology in cardiotoxicity evaluation, has emerged as an effective solution.

### 2.1. Primary Cardiomyocytes

Primary cardiomyocytes, such as rat cardiac ventricular myocytes (NRVM), are the oldest and useful model to study various aspects of cardiac function and pathologies. These cells are particularly interesting to investigate the effects of drugs on the heart, including cardiotoxicity induced by tyrosine kinase inhibitors (TKIs) and doxorubicin [31,32]. The adult cardiomyocytes are a more powerful tool for heart research because they are physiologically relevant and can accurately represent the functioning of the heart. However, isolating adult cardiomyocytes can be a delicate process because they are highly sensitive to changes in their environment and can easily become damaged or die during the isolation process [33]. On the other hand, human cardiomyocytes provide a more accurate model of drug-induced cardiotoxicities compared to animal cardiomyocytes, and this is because there are significant differences in the structure and function of the heart between humans and animals, and the response to drugs can vary greatly [34,35,36] Human cardiomyocytes are capable of retaining their morphological and electrophysiological properties for a short time when kept in culture, allowing for the study of drug effects on the heart in a controlled environment [37]. For example, some studies have shown that doxorubicin can also affect the contractile function and oxidative stress status of adult rat cardiomyocytes, which can further contribute to cardiac dysfunction [38,39,40].

### 2.2. Established Cell Lines

H9C2 cardiomyoblast cells have been isolated from the cardiac ventricle of embryonic rats and show many similarities to primary cardiomyocytes. These cells have been used to investigate the metabolism of drugs by cardiac enzymes, to study drug-induced toxicities and transmembrane signal transduction, and to validate the model for ischemia-reperfusion injury [41,42,43,44]. However, H9C2 cells also have characteristics of multiple skeletal muscles and can differentiate into myotube-like structures, which limits their use as a model for studying specific cardiac functions. It is important to note that while H9C2 cells share many similarities with primary cardiomyocytes, they are not a perfect substitute for these cells [41,43,45].

Mouse atrial HL-1 cells are immortalized cells with a cardiac phenotype. These cell lines have been widely used for in vitro studies relating to cardiovascular pathologies, less for determining the cardiotoxicity induced by antineoplastic molecules [46,47,48] Kuzenetsov et al., have demonstrated that H9C2 cells, which are also derived from rat heart tissue, are more similar to primary cardiomyocytes than HL-1 cells and therefore may be a better model for studying cardiotoxicity induced by antineoplastic molecules [44].

AC16 is a cell line that was derived from primary cells of human ventricular tissue [49]. This cell line has been extensively used to study the cardiotoxic side effects of various drugs; in particular, researchers have used the AC16 cell line to study the cumulative dose-dependent cardiotoxic effects of doxorubicin [50]. These cells overexpress hERG but do not have the electrophysiological and functional properties of cardiomyocytes, therefore, at the moment, no study has analyzed the cardiotoxicity induced by antineoplastics through this system.

### 2.3. Human Pluripotent Stem Cell-Derived Cardiomyocytes

The use of hiPSC-CMs (human induced pluripotent stem cell-derived cardiomyocytes) in cardio-oncology is a promising approach for evaluating the cardiotoxicity of anticancer drugs and predicting the risk of developing cardiac dysfunction in cancer patients. By using patient-specific hiPSC-CMs, it is possible to model the effects of anticancer drugs on the heart of individual patients and identify those who are at higher risk of developing cardiac dysfunction.

These cells are specific to the donor patient and with unlimited regeneration potential [51]. HiPSC-CMs have calcium flux, retain contractile function, and express most of the ion channels and sarcomeric proteins found in the adult heart [52]. This model was used to evaluate cardiotoxicity induced by both doxorubicin and erlotinib, a Tyrosine Kinase Inhibitor (TKI). It has been shown that doxorubicin is much more cardiotoxic than erlotinib and that each TKI has a different side effect on hiPSC-CMs [53]. Several studies have been performed to determine the effects of trastuzumab on the properties of iPSC-CMs isolated from tissues of healthy and diseased patients and it has been found that patients with severe cardiac dysfunction are more vulnerable to trastuzumab. However, as mentioned, hiPSC-CMs have some limitations, including their immature state, structural and electrophysiological variability, and the cost and time required for their extraction and isolation. Therefore, it is essential to continue improving the differentiation protocols to generate more mature and uniform hiPSC-CMs and to optimize the culture conditions to maintain their properties over time [54]. Moreover, it is necessary to validate the results obtained using hiPSC-CMs in animal models and clinical trials to ensure their predictive value and clinical relevance. This validation process will help to establish the safety and efficacy of using hiPSC-CMs in cardio-oncology and accelerate the translation of this technology into clinical practice. Overall, hiPSC-CMs represent a valuable tool for advancing our understanding of the cardiotoxicity of anticancer drugs and improving the management of cancer patients [54]. The different preclinical cellular models are summarized in Table 1.

## 3. Preclinical Animal Model

Preclinical animal models have played a critical role in understanding the underlying mechanisms of RIHD, and when designing studies to mimic a clinical scenario, it is important to carefully consider various parameters—such as the animal’s age, size, and gender—as these factors can significantly influence the experimental findings. The age of the animals used in the study is an important factor to consider because the response to radiation exposure may differ depending on the developmental stage of the animal. For example, younger animals may be more susceptible to radiation-induced damage, while older animals may be less sensitive. The size of the animal can also have an impact on the experimental results, particularly when it comes to the radiation dose. Larger animals may require a higher dose of radiation to induce similar effects as in smaller animals. Gender can also play a role in the response to radiation exposure, particularly when it comes to the incidence and severity of certain diseases. For example, males may be more susceptible to developing radiation-induced heart disease than females. Therefore, when designing studies to mimic a clinical scenario of RIHD, researchers should carefully consider these parameters to ensure that the experimental findings are as relevant and applicable to human patients as possible [29,64]

Zebrafish (*Danio rerio*) has emerged as a widely used model organism for investigating diverse biological processes due to its unique combination of advantages. These include its genetic similarity to humans, rapid development, optical transparency during early stages, and cost-effective maintenance. Zebrafish is particularly valuable in the fields of drug discovery and toxicity studies, including cardiotoxicity assessments, where its physiological and molecular responses closely resemble those of mammals. High-throughput screening using zebrafish enables the evaluation of potential therapeutic compounds and the identification of adverse drug effects in a time-efficient and scalable manner. Its versatility makes zebrafish a critical tool for advancing our understanding of complex biological mechanisms and improving preclinical research outcomes [65,66,67].

Mice and rats are the most commonly used preclinical animal models for RIHD research, mainly due to their small size, low maintenance costs, and ease of genetic manipulation. These animal models have been instrumental in identifying the pathways involved in the development of RIHD, as well as potential therapeutic targets [24,68,69,70]. Genetically engineered models of rodents have also been used in RIHD research. These animal models have specific mutations that predispose them to heart disease, such as atherosclerosis, and they are used to investigate how radiation affects the development and progression of pre-existing heart disease [71,72,73,74]. Inflammatory and thrombotic responses were found to be accelerated in atherosclerosis-prone ApoE (−/−) mice compared to wild-type C57BL/6J mice following ionizing radiation. Specific endothelial knockout mouse models of p53 and p21 were used to elucidate the role of endothelial cells in post-radiation cardiac injury [73,75,76,77,78]. Partial cardiac irradiation studies have also helped in understanding the molecular pathways that regulate the RIHD environment. However, rodent models have limitations due to their phylogenetic distance from humans and differences in their physiological responses to treatment patterns. Thus, there is a need for larger animal models to study RIHD [25,79,80,81,82].

Rabbits have been used as an animal model for studying cardiac failure, ischemic heart disease, and electrophysiological phenomena caused by radiation. Their physiological characteristics, including similarities in heart function, vascular structure, and blood flow, make them a valuable model for studying cardiovascular disease and the effects of radiation on the heart [26,83,84,85]. However, it is important to note that rabbits differ from mice in heart size, heart rate, and body weight, among other factors. These differences need to be taken into account, especially in studies of arrhythmia as a side effect of radiotherapy, to ensure that the results are applicable to humans [26,86]. Despite these differences, the extensive use of rabbits in cardiac radiation research since 1968 has led to the development of a large number of transgenic rabbit models of cardiovascular disease and anti-rabbit antibodies, facilitating their use in research [87,88,89].

Canine models offer several advantages, including similarities in organ and cellular characteristics with humans and similarities in the coronary circulation with elderly people with ischemic heart disease [90,91,92]. However, the limited use of canine models in cardiac radiation studies can be attributed to several factors. First, there are stricter regulations and approval procedures for the use of dogs in research, which can make it more difficult to obtain permission to use them in studies. Second, the high maintenance costs associated with caring for dogs can also be a barrier, particularly for smaller research groups with limited funding. Finally, the strong emotional bond between humans and dogs can create controversy and ethical concerns around their use in research [27,93,94].

The effects of RIHD are very common among pigs and nonhuman primates, but these species have been used by only a few groups and are more costly than other models [28,95,96].

The different preclinical animal models are summarized in Table 2.

### 3.1. Radiotherapy

#### Mechanisms of Radiation-Induced Cardiovascular Toxicity

One of the known consequences of radiation exposure to the heart is the development of Radiation-Induced Myocardial Fibrosis (RIMF), a chronic condition that may develop years after radiation therapy, and there is no known cure for it. Myocardial fibrosis refers to the buildup of excess collagen in the heart muscle tissue, which can cause the heart muscle to become stiff and less flexible. This can result in reduced heart function and lead to various symptoms, such as shortness of breath, fatigue, and chest pain [102,103,104].

The exact molecular mechanism underlying RIMF Is not fully understood, but it is known that the deposition of collagen in damaged heart tissue leads to an increase in myocardial thickness and a decrease in systolic and diastolic function [105]. One of the factors that contribute to the increased myocardial thickness is the presence of actin stress fibers in the collagen-producing myofibroblasts. Additionally, an activated inflammatory environment can also contribute to the development of RIMF [106,107]. This radiation side effect can lead to downstream diseases such as arrhythmia and cardiomyopathy through several mechanisms. For example, fibrosis affecting the cardiac conduction system can interfere with the transmission of electrophysiological signals, leading to arrhythmia [108]. Microvascular damage is also mechanistically linked to cardiomyopathy following radiotherapy, as reduction of capillaries supporting cardiomyocytes can lead to hypoxia and death of myocardial tissue with progressive fibrosis. Overall, RIMF is a serious complication of radiation therapy, and further efforts are needed to fully understand the molecular mechanisms underlying its development and to identify new effective treatments [109].

The development of RIMF involves complex cellular and molecular mechanisms, which include the induction of oxidative stress and inflammation [110]. Oxidative stress occurs when there is an imbalance between the production of reactive oxygen species (ROS) and the ability of the body’s antioxidant defense system to neutralize them. Radiation exposure can cause the production of ROS, which can damage cellular components such as proteins, lipids, and DNA. This damage can lead to the activation of signaling pathways that contribute to the development of RIMF. In addition to oxidative stress, radiation exposure can also lead to the priming of a pro-inflammatory environment in the heart. This involves the activation of immune cells, such as neutrophils and lymphocytes, which can release cytokines and growth factors that promote inflammation and tissue damage [111,112,113]. These cytokines and growth factors, including TGF-beta, TNF-alpha, IL-1, and IL-11, can attract more immune cells to the damaged heart, leading to a vicious cycle of inflammation and tissue damage. The inflammatory cascade starts with vascular damage and endothelial dysfunction, which can occur within hours of radiation exposure. This can lead to the adhesion of neutrophils to the endothelium and their migration to the damaged heart tissue. Over time, this can lead to the infiltration of more immune cells, leading to tissue fibrosis and impaired heart function [114,115,116]. Furthermore, the monocytes differentiate into a specific subtype of macrophages called M2 macrophages, which secrete a signaling molecule called TGF-beta. TGF-beta then induces the differentiation of fibroblasts and bone marrow progenitor cells into myofibroblasts, which are cells that are known to contribute to tissue fibrosis. Subsequently, myofibroblasts increase the synthesis of extracellular matrix (ECM), which is a complex mixture of molecules that make up the structural scaffold of tissues. This increased synthesis of ECM is accompanied by increased expression of integrins, which are cell-surface receptors that interact with the ECM, as well as tissue inhibitors of matrix metalloproteins (MMPs), which are enzymes that break down the ECM. Overall, these events contribute to the development of fibrosis in the heart tissue, which is a hallmark of RIHD (Figure 3) [117,118,119]. It is worth noting that fibrosis can impair heart function by stiffening the heart tissue and reducing its ability to pump blood efficiently, which can lead to symptoms such as shortness of breath, fatigue, and chest pain. The decrease in cardiac contraction may be triggered by coupled mechanoelectrical damage of cardiomyocytes caused by excess ECM. Furthermore, the deprivation of oxygen and nutrients due to inflammation and fibrosis can further aggravate cardiac remodeling and compromise elasticity and distensibility, favoring cardiac failure [116,120].

Finally, radiation-induced DNA damage can lead to the upregulation of the BAX protein and deregulation of the BCL2 protein in cardiomyocytes. BAX is a pro-apoptotic protein that promotes cell death, while BCL2 is an anti-apoptotic protein that helps to prevent cell death. The upregulation of BAX and deregulation of BCL2 can result in the apoptosis (programmed cell death) of cardiomyocytes [121]. Apoptosis of cardiomyocytes can lead to the development of fibrosis, which is the formation of excess connective tissue in the heart. Fibrosis can impair heart function, leading to the development of radiation-induced heart disease (RIHD). In summary, radiation-induced DNA damage can lead to the upregulation of BAX and deregulation of BCL2 in cardiomyocytes, resulting in apoptosis, fibrosis, and ultimately, the progression of RIHD [122,123].

The different mechanisms of radiation-induced cardiac toxicity are summarized in Table 3.

### 3.2. Chemotherapy

#### Mechanisms of Chemotherapy-Induced Cardiac-Toxicity

Cancer therapy can induce myocardial damage through various mechanisms, with each anticancer agent having unique effects on the heart. Anthracyclines, such as doxorubicin and epirubicin, can cause toxicity by oxidative stress, DNA damage, and inhibition of DNA repair [138]. Fluoropyrimidines such as 5-fluorouracil (5-FU) and capecitabine inhibit pyrimidine nucleotide biosynthesis and may lead to cardiotoxicity in about 30% of patients [139,140]. Alkylating agents such as cyclophosphamide and ifosfamide impact DNA transcription and protein synthesis, resulting in cardiac damage, especially with higher cumulative doses [141]. Microtubular polymerization inhibitors such as paclitaxel and docetaxel can increase the risk of heart failure in patients receiving anthracycline therapy [142,143,144,145]. Anti-HER2 therapy with trastuzumab can lead to cardiac toxicity, especially when combined with anthracyclines [146,147,148]. Vascular endothelial growth factor (VEGF) inhibitors may cause hypertension and atherosclerosis due to their effects on angiogenesis [149,150,151,152]. These various mechanisms contribute to the development of radiation-induced cardiovascular toxicity and require further understanding and research for effective treatment and management.

The different mechanisms of chemotherapy-induced cardiac toxicity are summarized in Table 4.

### 3.3. Immunotherapy

Immunotherapy has revolutionized cancer treatment by leveraging the power of the patient’s immune system to fight cancer. PD-1 is a protein expressed on the surface of T-cells that plays a critical role in regulating the immune response by suppressing T-cell activation and preventing autoimmunity. PD-1 interacts with its ligands PD-L1 and PD-L2, which are expressed on the surface of cancer cells and other cells in the tumor microenvironment. PD-1/PD-L1 inhibitors block the interaction between PD-1 and PD-L1, which releases the brakes on the immune system, allowing T-cells to recognize and attack cancer cells. This leads to an immune-mediated response against cancer cells and improved patient outcomes [182]. However, the activation of the immune system can also lead to immune-related Adverse Events (irAEs), which can affect various organs, including the heart. The exact mechanism of ICI-associated cardiotoxicity is not fully understood [183]. Proposed mechanisms include direct destruction of cardiac tissue by deregulated, activated autoimmune T lymphocytes, indirect destruction by pro-inflammatory cytokines released by ICI-deregulated T lymphocytes and activated cells, and recognition of cardiac self-antigens by autoantibodies promoting cell-mediated cardiotoxicity [184]. PD-1/PD-L1 inhibitors, such as pembrolizumab, can disrupt the immunologic homeostasis of cardiac structures mediated by CTLA-4 and PD-1/PD-L1 pathways, leading to autoimmune cardiac toxicity (Figure 4) [185,186].

Additionally, ICIs may induce cellular infiltration of cardiac myocytes through immune polarization effects [187]. Another theory suggests the existence of common T-cell receptors or epitopes between cardiac myocytes and tumors [188,189]. ICI-induced cytokine release, such as IL-6, IFN-g, TNFa, NO, NOS, and ROS, can have cytotoxic effects on cardiac myocytes, leading to various cardiac anomalies [190,191]. Dysregulation of myocardial metabolism, including alterations in lipid and glucose metabolism, oxidative phosphorylation, and mitochondrial function, may also contribute to myocardial dysfunction [192,193]. Further research is needed to fully understand these mechanisms and improve the management of ICI-induced cardiotoxicity [194].

The different mechanisms of immunotherapy-related cardiac toxicity are summarized in Table 5.

## 4. Examples of Application of In Vitro and In Vivo Models

Preclinical research plays a critical role in understanding the biological mechanisms of cancer therapies, optimizing treatment protocols, and developing strategies to mitigate therapy-induced side effects. This section introduces examples of in vitro and in vivo models used to investigate the effects of radiotherapy, chemotherapy, and immunotherapy on cancer and normal tissues. These models, designed to simulate the complex interplay of therapeutic agents with biological systems, have significantly advanced our knowledge of therapeutic efficacy and toxicity. Clinical studies serve as the critical bridge between preclinical findings and patient outcomes. They validate and contextualize preclinical data, offering insights into how therapies perform in diverse and complex human populations. However, the translation of preclinical research to clinical settings remains a significant challenge. Differences in drug metabolism, immune response, and tumor microenvironments between preclinical models and humans often limit the applicability of findings. Addressing this translational gap requires refining preclinical models to better mimic human biology and fostering closer integration between experimental research and clinical practice.

### 4.1. Examples of In Vitro and In Vivo Models to Study the Effects of Radiotherapy

In vitro models are valuable tools for studying the mechanisms of radiotherapy, evaluating the efficacy of radiation sensitizers or protectors, and developing novel treatment strategies for cancer and other diseases. They provide a controlled and reproducible environment for investigating the effects of radiation on biological systems.

The work of Kim et al. discusses the effects of high-dose irradiation on human induced pluripotent stem cell-derived cardiomyocytes (iPSC-CMs) and its potential implications for cardiac radioablation. iPSC-CMs were exposed to different doses of irradiation, ranging from 0 Gy (no irradiation) to 50 Gy, using a multi-electrode array to measure the electrical activities of these cells. After irradiation, iPSC-CMs exhibited changes in their electrophysiological activities. These changes included an increase in the beat rate immediately after irradiation, which peaked at 3 h but steadily decreased afterward. Additionally, conduction velocity slowed in cells irradiated with 25 Gy or more, but it recovered within 24 h. However, iPSC-CMs appeared to recover their electrophysiological activities, including a total active electrode, spike amplitude, and slope, and corrected field potential duration within 3 to 6 h from the acute effects of high-dose irradiation. These findings suggest that high-dose irradiation immediately and reversibly modifies the electrical conduction of cardiomyocytes. The changes observed in conduction velocity may be indicative of compensatory mechanisms activated at the cellular level after the acute effects of high-dose irradiation [194].

In radiation oncology research, animal models are widely used to study the effects of radiation therapy on various aspects of the body, including the cardiovascular system. In this context, the aim is to deliver a precise dose of radiation to target cancer cells while minimizing damage to surrounding healthy tissues. However, radiation can still affect nearby organs, including the heart and blood vessels. Animal models allow for the study of the effects of radiation on the cardiovascular system in a controlled and systematic manner. These models are selected based on their anatomical and physiological similarities to humans, allowing for meaningful extrapolation of data [25,28]. The effects of radiation on the heart and blood vessels in animal models can be evaluated through various methods such as echocardiography, electrocardiography, and invasive measurements. In addition, it is possible to analyze changes in biomarkers associated with cardiotoxicity, such as troponin levels, oxidative stress markers, and inflammatory markers [29].

In this context, a recent study by Rosen et al. focused on understanding the impact of radiation exposure on the heart, using spontaneously hypertensive Wistar-Kyoto rats as an animal model. These rats have previously been shown to develop drug-induced cardiomyopathy, making them a suitable model for studying radiation-induced heart damage. This study evaluated the effects of gamma irradiation at various time points, two weeks, four weeks, and fifty-two weeks after exposure. This approach allows for the assessment of both acute and long-term consequences of radiation exposure on the heart. In more detail, both male and female rats exposed to radiation experienced weight loss and anemia over a one-year period and low red blood cell counts. Elevated levels of the inflammatory marker IL-6 were detected in males at four weeks post-irradiation. In addition, serum analysis of cardiac troponin T and I, which are markers of heart damage, indicated signs of cardiomyopathy at earlier time points post-irradiation; however, there was high variability in these markers, particularly at the one-year mark. Echocardiography performed at two weeks post-irradiation revealed significant changes in cardiac function. Females showed a significant decrease in cardiac output, while males exhibited decreases in both diastolic and systolic volumes. All these results suggest impaired heart function [212].

In the same study, a separate experiment using normotensive Wistar-Kyoto rats exposed to higher radiation doses (10.0 Gy), heart tissue showed an increase in total protein oxidative carbonylation, indicating oxidative damage. DNA damage was also observed, as indicated by increased levels of γ-H2AX. By a proteomic analysis several proteins were identified with notable differences in carbonylation, including proteins of mitochondrial origin and cardiac troponin T. Carbonylation of cardiac troponin T is particularly significant, as it plays a crucial role in cardiomyocyte contractility. Overall, the study conducted by Rosen provides evidence of acute oxidative protein damage, DNA damage, carbonylation of cardiac troponin T, and the development of long-term cardiomyopathy in animals exposed to radiation. These findings highlight the potential risks associated with radiation exposure on heart health and suggest that oxidative damage and DNA damage may play a role in radiation-induced cardiomyopathy. The knowledge of these biological effects is crucial for improving the safety of radiation therapy and managing the cardiac consequences of radiation exposure in emergency situations [212].

In 2022, Yi et al. investigated the potential synergistic cardiotoxic effects of combining chest radiotherapy and targeted therapy with trastuzumab (TRZ) in breast cancer patients with overexpression of human epidermal growth factor receptor 2 (HER2). Breast cancer patients with HER2 overexpression often receive TRZ as part of their targeted therapy. However, the combination of TRZ and chest radiotherapy can potentially lead to cardiotoxicity, limiting the clinical benefits of treatment. The study established an in vitro model using H9C2 cardiomyocytes to assess the effects of TRZ and radiation on cardiomyocyte injury. They used various assays and tests, including cell flow cytometry, the CCK-8 test, Western blot, γ-H2AX fluorescence focus formation, and measurement of reactive oxygen species (ROS) content. The results indicated that concurrent TRZ treatment exacerbated the injury effect of irradiation on cardiomyocytes in vitro. This effect appeared to be linked to several factors, including the inhibition of Akt phosphorylation, the promotion of ROS accumulation in cells, and the induction of intracellular DNA damage. The study also established a mouse heart injury model by subjecting the animals to X-ray cardiac irradiation combined with TRZ treatment. They assessed cardiac function using small animal ultrasound and 18FDG-microPET/CT six months later. Additionally, they examined the morphological changes in heart tissue through histological sections. By in vivo experiments, the researchers observed dysfunction in diastolic function and myocardial ischemia in mouse hearts as measured by echocardiography and 18FDG-microPET/CT; they also observed myocardial fibrosis and cardiomyocyte apoptosis. In vitro and in vivo experiments demonstrated that the combination of TRZ and irradiation resulted in more significant cardiotoxicity than either treatment alone. These findings emphasize the importance of carefully considering the concurrent use of TRZ and radiotherapy in clinical practice, with a particular focus on cardiac safety [213].

#### Clinical Reports on Radiotherapy Effects

The application of translational research to radiation-induced cardiotoxicity faces significant challenges, primarily due to the complexities of clinical radiotherapy delivery for chest radiation [214,215,216]. Clinical radiotherapy protocols often involve long courses of treatment and significant advancements in precision, such as image-guided radiotherapy (IGRT), intensity-modulated radiation therapy (IMRT), volumetric modulated arc therapy (VMAT), and stereotactic body radiotherapy (SBRT) [217]. These techniques allow for substantial sparing of organs at risk (OARs) while delivering high doses to target tissues.

In contrast, animal models generally utilize less sophisticated radiotherapy delivery methods [29]. A predominant approach involves exposing the entire heart to high doses in single fractions, which poorly replicates the intricacies of clinical radiotherapy. Although recent developments in preclinical radiation techniques have enabled improved dose delivery and fractionation in small animals, the quality of radiotherapy in animal studies still lags behind clinical standards [218]. These differences limit the direct applicability of preclinical findings to clinical scenarios, particularly in understanding long-term, fractionated radiotherapy effects on cardiac structures.

Another limitation lies in the inherent differences between animal models and the patient populations undergoing chest radiotherapy. Clinical patients are typically older and may present with pre-existing comorbidities, such as cardiovascular and pulmonary diseases, which significantly influence the side effects of radiation on the heart [219]. These underlying conditions, coupled with patient-specific factors such as age, sex, and lifestyle, create a complex environment that is difficult to replicate in animal models.

While recent preclinical studies have made strides toward mimicking these factors—incorporating aged and sex-specific animal models—their ability to fully capture the nuances of human populations remains limited [220]. For instance, young, healthy animals used in traditional models fail to replicate the compounded impact of co-pathologies or the variations in physiological responses observed in clinical populations [29].

Despite these limitations, translational research may provide meaningful contributions, particularly in the identification of sensitive biomarkers for early detection and risk stratification of radiation-induced cardiotoxicity [29,221]. These biomarkers, which include both serum markers and imaging-based indicators, aim to diagnose cardiotoxicity before tissue damage becomes irreversible. Such advancements hold promise for enabling more targeted and effective therapeutic interventions in both preclinical and clinical settings.

Preclinical studies have also shed light on the dose-response relationships and histologic changes in cardiac tissues. For example, SBRT studies on normal cardiac conduction systems in animal models have provided foundational knowledge [222]. Research has demonstrated that a dose of 25 Gy SBRT to specific cardiac regions, such as the pulmonary vein and atrioventricular node, induces conduction blockade without immediate toxicity. Lehmann et al., reported a median time to complete an atrioventricular block of 11 weeks in large animal models, underscoring the feasibility of applying radiotherapy to modulate electrical conduction while minimizing short-term adverse effects [223].

Advances in imaging, radiation delivery, and dosimetry have created opportunities to improve preclinical models and better mimic clinical radiotherapy. By refining experimental setups and dose distributions, researchers can gain a deeper understanding of normal tissue radiation responses. However, rigorous radiation dosimetry and detailed reporting of preclinical experimental setups are critical to ensuring reproducibility across studies. Finally, recognizing and addressing the differences in thoracic radiotherapy protocols across research laboratories will enhance the interpretability of preclinical findings. This harmonization is essential for elucidating the mechanisms of radiation-induced cardiopulmonary damage and translating preclinical insights into improved clinical outcomes.

### 4.2. Examples of In Vitro and In Vivo Models to Study the Effects of Chemotherapy

Doxorubicin, a well-known chemotherapy drug, is often related to cardiotoxic effects, which limits its use in cancer treatment. Dexrazoxane is currently the only approved drug for partially protecting against doxorubicin-induced cardiotoxicity, but its use is typically restricted to patients receiving a high cumulative dose of anthracyclines. The study of Tomlinson et al. aims to address the limited availability of relevant in vitro human cardiac model systems for investigating doxorubicin-induced cardiotoxicity and potential cardioprotective strategies. Researchers adapted a functional 3D human multi-cell type cardiac system to mimic patient responses to doxorubicin and dexrazoxane. Two NRF2 gene inducers, Bardoxolone methyl (a semi-synthetic triterpenoid) and sulforaphane (an isothiocyanate) were administered to the cardiac model system. Data collected showed that these NRF2 gene inducers provided cardioprotection against doxorubicin toxicity, and this protection was comparable to the effects of dexrazoxane. Cardioprotection was evidenced by an increase in cell viability and a decrease in the production of reactive oxygen species (ROS). Additionally, when the NRF2 gene inducers were used in combination with dexrazoxane, a synergistic reduction in cardiotoxicity was observed. The study highlights the importance of the NRF2 pathway in cardioprotection and suggests that these NRF2 gene inducers could serve as a novel pharmacological intervention to alleviate doxorubicin-induced cardiotoxicity [224].

In cancer research, xenograft models are commonly used. These models involve implanting human tumor cells or tissue into immunodeficient mice, and they are allowed to evaluate the response of tumors to chemotherapy drugs and to monitor key factors such as tumor growth inhibition, regression, and metastasis. Animal models also provide a means to study the systemic effects of chemotherapy on various organs and tissues. Although animal models have been invaluable in chemotherapy research, it is essential to acknowledge their limitations. Differences in drug metabolism, immune response, and tumor microenvironment between animals and humans can impact the translation of findings. Therefore, findings from animal studies should be interpreted cautiously and further validated in clinical trials.

Rodent models, particularly Wistar-Kyoto mice, and DBA/2J mice, have been commonly used to study anthracycline-induced cardiotoxicity, which is a serious side effect of chemotherapy drugs such as doxorubicin. These models have shown that hypertension and gender can impact the susceptibility to cardiotoxicity [225,226]. For example, Wistar-Kyoto mice spontaneously expressing hypertension are more sensitive to doxorubicin-induced cardiotoxicity than normotensive mice, and male rodents are more susceptible than females. In the case of Wistar rats, doxorubicin-induced toxicity was found to differ between adult male and female rats. Male rats showed evident signs of cardiomyopathy, cardiac atrophy, and reduced left ventricular ejection fraction, leading to a high mortality rate of 50%, while female rats showed only a moderate reduction of left ventricular ejection [227,228]. Tumor-type xenograft models have also been used to evaluate the cardiotoxic effects of doxorubicin, and similar gender differences have been observed. Male xenograft models were found to develop higher cardiotoxicity than female models [229].

Zebrafish cardiomyocytes express vertebrate-like sodium, calcium, and potassium channels useful for studying drug-associated QT prolongation [230,231]. Furthermore, the development of genome editing technologies such as CRISPR/Cas9 has allowed researchers to create transgenic zebrafish models that are specifically tailored to study cardiotoxicity. It has been observed that zebrafish embryos exposed to anthracycline, such as daunorubicin, pirarubicin, and doxorubicin, developed heart defects. Furthermore, the researchers have created zebrafish models with mutations in the CYP1a gene, which protects them from doxorubicin-induced cardiotoxicity. Such models can provide valuable insights into the mechanisms underlying drug-induced cardiotoxicity and may lead to the development of new treatments for this condition [232].

In particular, Legi et al., have investigated the role of Substance P (SP), a neuropeptide, and its high-affinity receptor, NK-1R, in chemotherapy-associated cardiotoxicity induced by Doxorubicin (DOX) in mice. C57BL/6 mice were used for the study and were divided into several groups: DOX-treated, DOX-treated with different dosages of aprepitant (NK-1R antagonist), and control groups. DOX was administered intraperitoneally once a week for five weeks. Aprepitant was administered in the drinking water at five different dosages. SP and NK1R levels were significantly increased in the hearts of mice treated with DOX compared to those without DOX treatment. DOX-induced cardiac dysfunction, as indicated by parameters such as fractional shortening (FS), ejection fraction (EF), and stroke volume (SV), was significantly attenuated by treatment with aprepitant. Lower doses of aprepitant effectively reduced the impact of DOX on these parameters, bringing them to levels such as control (vehicle-treated) mice. DOX treatment increased levels of cardiac apoptosis, oxidative stress, and cardiomyocyte hypertrophy, all of which were significantly reduced by treatment with lower doses of aprepitant in DOX-treated mice [233].

In 2022, Galán-Arriola et al. investigated the changes in coronary microcirculation during and after multiple cycles of anthracycline treatment. The study used large-white male pigs as an animal model. Pigs were divided into different experimental protocols based on the cumulative exposure to doxorubicin and the follow-up duration. The experimental groups included control (no doxorubicin), single doxorubicin injection with a sacrifice at 48 h or 2 weeks, three doxorubicin injections with a sacrifice at 2 weeks or 12 weeks, and five doxorubicin injections with a sacrifice at 8 weeks. Cardiac magnetic resonance (CMR) was used to quantify perfusion, and invasive measurements of coronary flow reserve (CFR) were performed to assess microcirculation. After each protocol, animals were sacrificed for ex vivo analyses, and vascular function was evaluated through myography in coronary arteries. A single doxorubicin injection had no immediate impact on microcirculation, ruling out direct chemical toxicity. However, a series of five doxorubicin injections with a high cumulative dose led to progressive and irreversible damage to microcirculation. This was evidenced by reduced myocardial perfusion and impaired functional microcirculation measured by CFR. Interestingly, even a low cumulative dose of doxorubicin (three injections) did not result in cardiac contractile deficits during long-term follow-up but did cause persistent microcirculation damage, apparent soon after the third dose injection. Histological and myograph evaluations confirmed structural damage to arteries of all sizes in animals undergoing low cumulative dose regimes. Arteriole damage and capillary bed alteration occurred only after high cumulative dose regimes [234].

#### Clinical Reports on Chemotherapy Effects

Translational research bridging preclinical findings to clinical applications in chemotherapy, particularly with anthracyclines, has led to significant advances in understanding and mitigating cancer therapy-related cardiovascular toxicity [235]. Preclinical studies have identified mitochondrial dysfunction as a primary mechanism of anthracycline-induced cardiotoxicity (AIC), with doxorubicin (DOX) binding to cardiolipin in the mitochondrial membrane, leading to increased oxidative stress, mitochondrial DNA damage, and impaired mitophagy [236]. This mechanistic understanding has translated into human studies, where strategies such as pharmacologic inhibition of phosphoinositide 3-kinase (PI3K)-γ have demonstrated potential in preserving cardiac function by promoting mitophagy [237].

Two major international research initiatives, the Leducq Foundation, and the European Commission-funded RESILIENCE project, have played a crucial role in advancing translational approaches to mitigate AIC [235]. The Leducq Foundation-funded consortium, Targeted Approaches for Prevention and Treatment of Anthracycline-Induced Cardiotoxicity, has been instrumental in elucidating molecular pathways involved in anthracycline toxicity [238]. Their research has identified PI3K signaling as a central modulator of mitochondrial dysfunction and cardiomyocyte senescence, leading to potential therapeutic targets aimed at enhancing mitophagy and reducing oxidative stress [238]. Meanwhile, the RESILIENCE project (Remote Ischemic Conditioning in Lymphoma Patients Receiving Anthracyclines) is evaluating the clinical application of remote ischemic conditioning (RIC), a non-invasive intervention that involves transient limb ischemia-reperfusion to activate cardioprotective pathways [237]. Preclinical studies have demonstrated that RIC preserves cardiac function by reducing anthracycline-induced oxidative stress and mitochondrial damage, and its ongoing clinical trial aims to establish its efficacy in preventing chemotherapy-induced heart failure [239].

Another key translational approach has involved refining preclinical models to better replicate human disease [235]. Traditional rodent models have limitations in mimicking human anthracycline cardiotoxicity, prompting the development of larger animal models such as pigs. These models have been instrumental in studying progressive left ventricular dysfunction and validating early imaging biomarkers such as T2 relaxation time prolongation in cardiac magnetic resonance imaging, which has been identified as an early marker of myocardial edema and damage in patients undergoing chemotherapy [240]. This has led to ongoing clinical trials aiming to implement CMR as a routine monitoring tool for early detection of cardiotoxicity in cancer patients [241].

Endothelial dysfunction and microvascular injury have also been highlighted as key contributors to AIC, with preclinical studies demonstrating that anthracyclines impair nitric oxide production and promote vascular inflammation [242]. These findings have been validated in clinical studies showing decreased myocardial perfusion in anthracycline-treated patients, reinforcing the need for cardioprotective interventions [243]. The collaborative efforts of Leducq and RESILIENCE are driving the integration of preclinical discoveries into clinical practice, ensuring that novel strategies for mitigating chemotherapy cardiotoxicity are evidence-based and widely applicable [235].

Collectively, these translational efforts underscore the importance of bidirectional integration between laboratory and clinical research. By leveraging preclinical insights into mitochondrial dysfunction, endothelial injury, and oxidative stress, human studies have advanced diagnostic and therapeutic approaches, paving the way for personalized cardio-oncology care and improved long-term outcomes for cancer patients receiving chemotherapy.

### 4.3. Examples of In Vitro and In Vivo Models to Study the Effects of Immunotherapy

Ipilimumab is a monoclonal antibody that targets the protein CTLA-4 and enhances T-cell activation, and it is used to treat advanced melanoma. Studies have shown that treatment with ipilimumab can cause irAEs, including myocarditis and other cardiac complications, in some patients. Similarly, the combination of pembrolizumab, a PD-1 inhibitor, and trastuzumab, a HER2-targeted antibody used to treat breast cancer, has been associated with increased interleukin expression in female C57B1/6 mice [244,245].

Quagliariello et al., in 2019, investigated the potential cardiotoxic and pro-inflammatory effects of combining Pembrolizumab and Trastuzumab, two medications used in the treatment of cancer, particularly for HER2-positive breast cancer. The study used human fetal cardiomyocytes for in vitro experiments. Various tests were conducted, including cell viability assessment, measurement of intracellular calcium levels, and pro-inflammatory studies. Pro-inflammatory markers, including Interleukin 1β, 6, and 8, as well as the expression of NF-kB and Leukotriene B4, were analyzed. The results showed that the combination of Pembrolizumab and Trastuzumab significantly increased intracellular calcium overload, reaching three times the levels of untreated cells. Cardiomyocyte viability was reduced by 65% compared to untreated cells. It was also reduced by 20–25% when compared to cells treated with Pembrolizumab or Trastuzumab alone. The combination therapy increased the inflammation of cardiomyocytes by enhancing the expression of NF-kB and Interleukins. This work highlights that the combination of Pembrolizumab and Trastuzumab has strong pro-inflammatory effects on the heart and that these effects are mediated by the overexpression of NF-kB and Leukotriene B4-related pathways [245].

Animal models, typically mice, are used to study various aspects of immunotherapy, including the development and testing of novel immunotherapeutic agents, combination therapies, and personalized treatment approaches. These models allow for the investigation of the interactions between the immune system, tumor cells, and the tumor microenvironment. One commonly used animal model in immunotherapy research is the humanized mouse model. These mice are engineered to possess a functional human immune system, allowing researchers to study the interactions between human immune cells and human tumor cells. Humanized mouse models provide insights into the immune response to immunotherapeutic agents, immune cell trafficking, and tumor-immune cell interactions [244]. Additionally, animal models enable the evaluation of potential side effects and toxicities associated with immunotherapies. For instance, researchers can monitor irAEs, such as autoimmune reactions, and assess their impact on various organs, including the heart, lungs, liver, and gastrointestinal tract. The immune systems of mice and humans differ in several aspects, and the tumor microenvironment may also vary. Therefore, findings from animal studies should be validated in clinical trials and human studies to ensure their relevance and applicability in human immunotherapy. Overall, animal models serve as critical tools for advancing our understanding of immunotherapy mechanisms, optimizing treatment strategies, and improving patient outcomes in the field of cancer immunotherapy [245].

The same group mentioned above, Quagliariello et al., also carried out a study about the cytotoxic and pro-inflammatory properties of Ipilimumab and Nivolumab, the underlying pathways, and the cytokine storm involved in their cardiotoxic effects. Co-cultures of human cardiomyocytes (heart muscle cells) and lymphocytes (a type of immune cell) were exposed to Ipilimumab or Nivolumab. The study assessed cell viability and measured the expression of various factors, including leukotrienes, NLRP3 (a protein involved in inflammation), MyD88 (a protein involved in immune signaling), and p65/NF-kB (a transcription factor associated with inflammation). In addition to in vitro experiments, C57 mice were treated with Ipilimumab, and their cardiac function was analyzed using 2D echocardiography before and after treatment. The expression of NLRP3, MyD88, p65/NF-kB, and several cytokines was analyzed in the heart tissue of the mice. Both Nivolumab and Ipilimumab exhibited effective anticancer properties but also had significant cardiotoxic effects in co-cultures of lymphocytes and either tumor or cardiac cells. Exposure to both ICIs increased the expression of NLRP3, MyD88, and p65/NF-kB in the co-culture experiments, indicating pro-inflammatory responses. However, Ipilimumab appeared to induce more pronounced pro-inflammatory and cardiotoxic effects compared to Nivolumab. Mice treated with Ipilimumab showed a significant decrease in fractional shortening and radial strain, indicating impaired cardiac function when compared to untreated mice. Analysis of myocardial tissue from treated mice revealed increased expression of NLRP3, MyD88, and several interleukins, further confirming pro-inflammatory responses in the heart. From these studies, it emerged that both Nivolumab and Ipilimumab exhibit cytotoxic effects mediated by the NLRP3 inflammasome and other inflammatory pathways [244].

#### Clinical Reports on Immunotherapy Effects

The translation of preclinical research on immune checkpoint inhibitor (ICI)-induced cardiotoxicity into clinical applications faces significant challenges due to limitations in existing models. Cellular models, such as induced pluripotent stem cell-derived cardiomyocytes (iPSC-CMs), are ineffective because they lack an adaptive immune system, which is essential for studying ICI-mediated toxicity [235]. Research using animal models, primarily murine, along with clinical data, has provided insights into the cellular mediators involved, as well as the pathogenic soluble factors, such as cytokines and chemokines, and the clonality and specificity of T cell receptors (TCR) [246].

Animal models, particularly mice, also have significant drawbacks. Wild-type mice rarely develop spontaneous ICI-induced myocarditis, requiring genetically modified strains that may not fully represent human disease. Additionally, mice have key differences in immune regulation, and their responses to ICIs often require artificially high doses, unlike the unpredictable toxicity seen in patients [247]. While ICIs, such as anti-PD-1 and anti-CTLA-4 therapies, have revolutionized cancer treatment, their nonspecific activation of the immune system can lead to immune-related adverse events (irAEs), including myocarditis, which, although rare (0.75–1.7%), has a high fatality rate (20–50%) [248]. Preclinical murine models have provided crucial mechanistic insights, demonstrating that ICI-induced myocarditis is primarily mediated by CD8+ T cells, with autoreactive T cells targeting cardiac antigens, such as α-myosin heavy chain (α-MyHC) [249]. However, these models often fail to fully recapitulate the human disease, as they typically involve genetically modified or immunocompromised mice that may not reflect the complexity of the human immune response. Additionally, the spontaneous development of ICI-induced cardiotoxicity in these models is rare, requiring artificial stimulation through repeated high-dose drug administration, which does not accurately mimic clinical dosing regimens [235].

Human studies have identified shared T-cell clones between tumors and myocardium in ICI-induced myocarditis patients, suggesting a role for molecular mimicry, where T cells recognizing tumor antigens may also cross-react with cardiac proteins [249]. However, translating these findings into effective risk stratification and therapeutic strategies remains challenging. Immunosuppressive therapy, primarily corticosteroids, has been the mainstay of treatment, but responses are variable, and some patients require additional interventions, such as intravenous immunoglobulins. The limitations of preclinical models necessitate the development of more representative systems, such as human-induced pluripotent stem cell (iPSC)-derived cardiomyocytes and organoid models, to better predict patient-specific immune responses [250]. These translational efforts are crucial for improving early detection, developing targeted immunomodulatory therapies, and mitigating cardiovascular complications associated with immunotherapy [235].

## 5. Practical Points

### 5.1. Collaboration and Selection of Animal Models

Establishing effective collaboration between clinicians and preclinical scientists is crucial for designing relevant experiments. Together, they should carefully select appropriate animal models that accurately mimic the cardiovascular complications observed in cancer patients. Considerations should be given to the species, strain, age, and sex of the animals to ensure the translatability of findings to the human context. For example, choosing animal models with cardiovascular similarities to humans will enhance the relevance and applicability of the study outcomes in the clinical setting.

### 5.2. Comprehensive Evaluation of Cardiac Function

Preclinical studies should encompass a comprehensive evaluation of cardiac function to identify potential cardiotoxic effects of cancer treatments. Utilizing a combination of cardiological biomarkers and advanced techniques, such as echocardiography, electrocardiography, advanced radiological imaging, and hemodynamic measurements, can provide valuable insights into the impact of cancer therapies on the heart. Monitoring changes in cardiac structure, function, and electrical activity will aid in early detection and assessment of cardiotoxicity, enabling timely interventions to safeguard patients’ cardiovascular health.

### 5.3. Mechanism Investigation and Validation

In vitro studies using cellular and molecular techniques play a critical role in investigating the underlying mechanisms of cardiotoxicity induced by radiation, chemotherapy, and immunotherapy. Conducting meticulous research on cellular and molecular pathways involved in the development of cardiac toxicity will contribute to a better understanding of the mechanisms driving these adverse effects. Additionally, researchers should validate the efficacy and safety of potential interventions aimed at mitigating cardiotoxicity using appropriate experimental designs and outcome measures. These validation steps are essential before considering the translation of preclinical findings into clinical applications.

By focusing on these three main points, clinicians, biologists, and preclinical scientists can collaborate effectively to design impactful preclinical studies for cardio-oncology endpoints. Combining expertise from different fields and utilizing relevant animal models will contribute to a comprehensive understanding of cardiotoxicities induced by different cancer treatments. This knowledge will ultimately aid in developing safer and more effective therapeutic strategies for cancer patients, minimizing the risk of cardiovascular complications and improving overall treatment outcomes.

## 6. Conclusions

In conclusion, despite significant advancements in preclinical research aimed at understanding and mitigating the cardiotoxic effects of various cancer therapies, several limitations persist, hindering the translational value of these findings. While in vitro and in vivo models have provided valuable insights into the mechanisms of cardiotoxicity induced by chemotherapy, radiotherapy, and immunotherapy, the gap between preclinical studies and clinical applications remains substantial. Current models often fail to fully replicate the complexity of human physiology, particularly in terms of immune responses, drug metabolism, and the multifactorial nature of cardiovascular diseases. Additionally, differences in species-specific responses, the lack of long-term follow-up in animal studies, and the inability to fully mimic the tumor microenvironment and patient comorbidities further limit the applicability of preclinical findings to human patients. These challenges underscore the need for more sophisticated and human-relevant models, such as patient-derived induced pluripotent stem cells (iPSCs), organ-on-a-chip technologies, and advanced genetically engineered animal models that better reflect human disease states. Future strategies should focus on integrating multi-omics approaches, refining experimental designs, and fostering closer collaboration between preclinical researchers and clinicians to bridge the translational gap. By addressing these limitations, the development of a reliable and predictive platform for assessing cardiotoxicity will not only enhance our understanding of the underlying mechanisms but also pave the way for safer and more effective cancer therapies, ultimately improving patient outcomes.

## Figures and Tables

**Figure 1 cancers-17-01059-f001:**
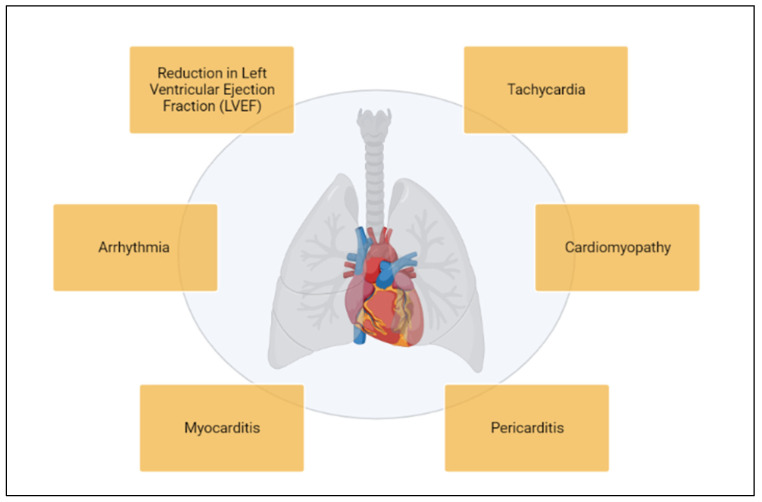
Anti-cancer treatments’ side effects on heart.

**Figure 2 cancers-17-01059-f002:**
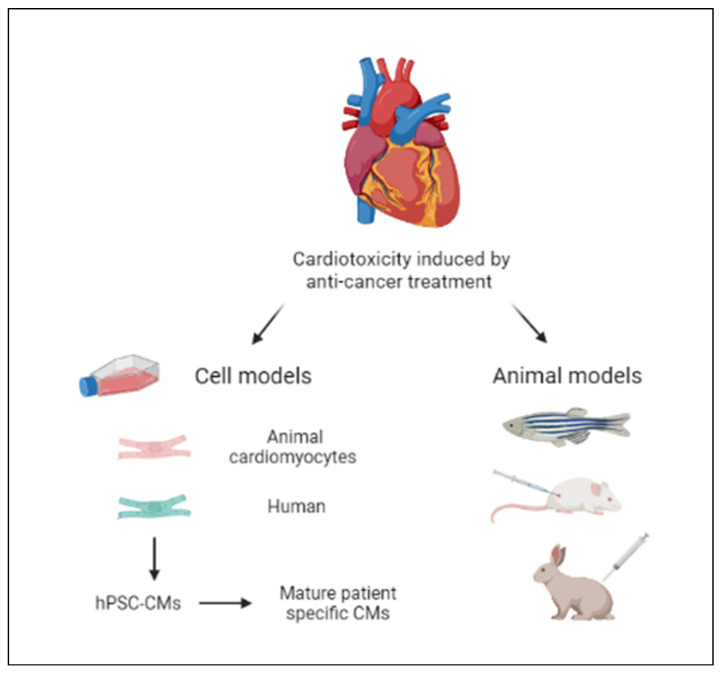
Research models for the evaluation of cardiotoxicity induced by anti-cancer treatments.

**Figure 3 cancers-17-01059-f003:**
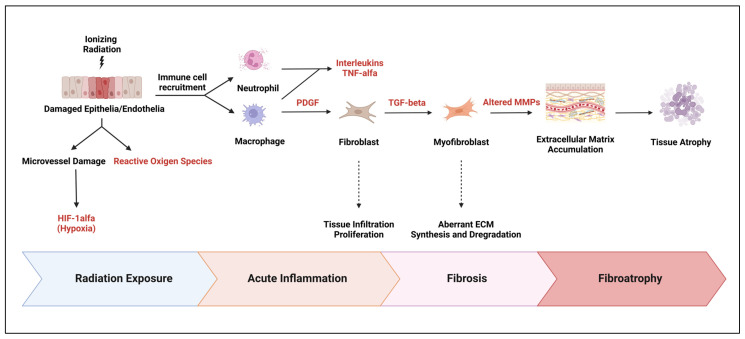
Pathway illustrating the progression from radiation-induced epithelial/endothelial damage to fibroatrophy via inflammation, fibroblast activation, ECM remodeling, and tissue atrophy.

**Figure 4 cancers-17-01059-f004:**
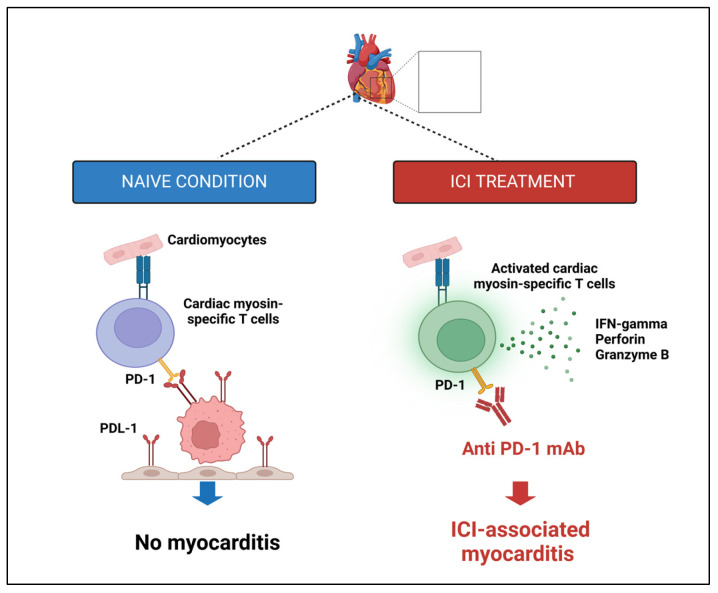
Diagram illustrating the development of ICI-associated myocarditis. Under naïve conditions, PD-1/PD-L1 interactions prevent T cell activation, avoiding myocarditis. With ICI treatment (anti-PD-1 mAb), cardiac myosin-specific T cells are activated, releasing IFN-γ, perforin, and granzyme B, leading to immune-mediated myocarditis.

**Table 1 cancers-17-01059-t001:** Summary of Studies Investigating Cardiotoxic Effects on Different Cell Types and Their Respective Cardiac Endpoints.

Cell Type	Results of the Study	References
Primary Cardiomyocytes	Doxorubicin impairs contractility, increases oxidative stress, and induces protein accumulation in cardiomyocytes, causing cardiac dysfunction	[38,39,40,55,56,57]
Established Cell Lines	H9C2 cells, resembling cardiomyocytes with skeletal traits, are a better model than HL-1 cells for studying antineoplastic cardiotoxicity	[41,43,44,45,58,59,60]
Human Pluripotent Stem Cells	Doxorubicin is more cardiotoxic than erlotinib; TKIs have varying effects, and severe cardiac dysfunction increases trastuzumab risk	[51,53,61,62,63]

**Table 2 cancers-17-01059-t002:** Overview of Animal Studies Investigating Radiation-Induced Heart Disease (RIHD) and Identifying Therapeutic Targets in Mice/Rats, Rabbits, Canines, and Pigs/Nonhuman Primates.

Animal	Description	References
Mice/Rats	Explored RIHD pathways, radiation-induced heart disease, and inflammation-related injury	[68,69,70,72,73,75,77,78,79,80,81]
Rabbits	Model for cardiovascular and radiation studies with distinct physiology from mice	[84,85,87,88,89,97,98]
Canines	Useful for cardiac radiation studies but limited by cost, regulations, and ethical concerns	[90,91,92,93,99]
Pigs/Nonhuman Primates	Common RIHD effects, but limited by higher costs	[94,95,96,100,101]

**Table 3 cancers-17-01059-t003:** Mechanisms of Radiation-Induced Cardiac Toxicity, Including Radiation-Induced Myocardial Fibrosis, Collagen and Actin Stress Fiber Deposition, Inflammation and Tissue Damage, and Fibrosis Leading to Impaired Heart Function.

Mechanism	Description	References
Radiation-Induced Myocardial Fibrosis (RIMF)	Chronic condition with excess collagen in heart tissue causes stiffening, reduced function, and symptoms such as shortness of breath, fatigue, and chest pain	[103,104,105,124,125,126]
Deposition of Collagen and Actin Stress Fibers	Increased myocardial thickness results from collagen deposition and actin stress fibers in collagen-producing myofibroblasts	[106,107,127,128,129,130]
Inflammation and Tissue Damage	Radiation exposure causes oxidative stress, inflammation, and immune cell activation, leading to tissue fibrosis and impaired heart function	[110,111,112,131,132,133,134,135]
Fibrosis and Impaired Heart Function	Myofibroblasts induce fibrosis, stiffening the heart and reducing efficiency, potentially causing cardiac failure	[117,118,119,136,137]

**Table 4 cancers-17-01059-t004:** Cardiotoxic Agents in Chemotherapy, Describing Anthracyclines, Fluoropyrimidines, Alkylating Agents, Microtubular Polymerization Inhibitors, Anti-HER2 Therapy, and VEGF Inhibitors, and Their Respective Mechanisms of Cardiotoxicity.

Agent	Description	References
Anthracyclines	Anthracyclines can cause cardiopathy through oxidative stress, DNA damage, and inhibition of DNA repair	[138,153,154,155,156]
Fluoropyrimidines	Fluoropyrimidines can cause cardiotoxicity in about 30% of patients by inhibiting pyrimidine nucleotide biosynthesis	[140,157,158,159,160,161]
Alkylating agents	Alkylating agents can cause cardiac damage, particularly at higher doses, by affecting DNA transcription and protein synthesis	[162,163,164,165]
Microtubular polymerization inhibitors	Microtubular polymerization inhibitors can increase the risk of heart failure in patients undergoing anthracycline therapy	[142,166,167,168,169,170]
Anti-HER2 therapy	Trastuzumab-based anti-HER2 therapy can cause cardiac toxicity, particularly when combined with anthracyclines	[146,147,148,171,172,173,174,175,176]
VEGF inhibitors	VEGF inhibitors can cause hypertension and atherosclerosis by affecting angiogenesis	[149,151,152,177,178,179,180,181]

**Table 5 cancers-17-01059-t005:** Mechanisms of Cardiotoxicity in Immunotherapy, Including Direct Cellular Destruction, Cardiac Antigen Immune Reactivity, Cytokine Release Syndrome, and Dysregulation of Myocardial Metabolism.

Mechanism	Description	References
Direct cellular destruction of cardiac tissue	ICI therapy disrupts cardiac immune balance, causing autoimmune toxicity with increased PD-L1 expression and T lymphocyte infiltration in myocarditis	[184,185,186,188,189,191,192,193,195,196,197]
Cardiac antigen immune reactivity	Myocarditis can result from disrupted molecular mimicry and autoantibodies induced by ICI therapy, leading to myocardial dysfunction	[188,189,198,199,200,201,202]
ICI-induced cytokines release	Therapies activating T-cell subsets can cause cytokine release syndrome (CRS), with pro-inflammatory cytokines and radicals damaging cardiac myocytes and causing cardiac anomalies	[190,203,204,205,206,207]
Dysregulation of myocardial metabolism	Anti-PD1 therapy can disrupt cardiomyocyte metabolism, affecting lipid/glucose metabolism mitochondrial function, and contributing to myocardial dysfunction and cell death	[192,193,194,208,209,210,211]

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
