# Peer review of "From Bench to Bedside: Translational Approaches to Cardiotoxicity in Breast Cancer, Lung Cancer, and Lymphoma Therapies"

_cancers, 2025, doi:10.3390/cancers17071059_

Round 1

Reviewer 1 Report

Comments and Suggestions for Authors

Nardone et al. reviewed translational approaches investigating cardiotoxicity induced by radiotherapy, chemotherapy, and immunotherapy. The topic is interesting and timely; however, the focus of this MS is not apparent. It is challenging to comprehensively review the clinical and preclinical research aspects of the cardiotoxic effects of radiotherapy, chemotherapies, and immunological therapies due to the different combinations and therapeutic regimens in various cancer types and stages. I suggest focusing on the most common therapeutic regimens of several common cancer types, e.g., breast cancer, NSCLC, and/or lymphomas. 

1. According to the Introduction, thoracic tumors, including lung cancer and lymphoma, as well as breast cancer, seem to be base of the selection of the reviewed oncologic treatments; however, it is not clear from the title and abstract. Please modify them accordingly or clearly state the base of your selection.

2. You mention that lung cancer patients have good survival rates. However, their 5-year survival is around 65% in cases of localized tumors and 8-25% in cases of distant metastases, which is far away from good survival. Breast cancer and childhood lymphoma patients show the best long-term survival rates among thoracic tumors; therefore, they have a higher chance of developing chronic cardiovascular complications. Please summarize the epidemiologic data of these tumors in the Introduction if you choose to focus on these tumor types.

3. What was the basis for selecting preclinical studies using animal models and pharmacologic interventions against cardiotoxicity? Some highlighted studies can not comprehensively review the broad literature on cardiotoxic mechanisms and potential interventional opportunities. 

4. The tables should not contain whole sentences because they are the repetition of the main text. They should be expanded with further important studies. Please provide the references in a separate column within the tables. It would be helpful to give the species/type of animal, sex, age, follow-up time, therapeutic intervention (with regimen, if applicable), and main molecular mechanisms with keywords in the tables.

5. Figures summarizing the molecular mechanisms and interactions of the different oncologic treatments are missing.

6. Beyond age, sex, and gender, comorbidities and prescribed drugs can modify the (side) effects of the oncologic treatments. Please mention these factors also.

Author Response

Reviewer 1

Nardone et al. reviewed translational approaches investigating cardiotoxicity induced by radiotherapy, chemotherapy, and immunotherapy. The topic is interesting and timely; however, the focus of this MS is not apparent. It is challenging to comprehensively review the clinical and preclinical research aspects of the cardiotoxic effects of radiotherapy, chemotherapies, and immunological therapies due to the different combinations and therapeutic regimens in various cancer types and stages. I suggest focusing on the most common therapeutic regimens of several common cancer types, e.g., breast cancer, NSCLC, and/or lymphomas. 

  1. According to the Introduction, thoracic tumors, including lung cancer and lymphoma, as well as breast cancer, seem to be base of the selection of the reviewed oncologic treatments; however, it is not clear from the title and abstract. Please modify them accordingly or clearly state the base of your selection.

Thank you for your valuable comment. In accordance with your suggestion, we have revised both the title and the abstract to clarify that our review focuses primarily on oncologic treatments for thoracic tumors, including lung cancer, lymphoma, and breast cancer. These modifications ensure greater consistency across the sections of the manuscript and explicitly reflect the basis of our selection.

  1. You mention that lung cancer patients have good survival rates. However, their 5-year survival is around 65% in cases of localized tumors and 8-25% in cases of distant metastases, which is far away from good survival. Breast cancer and childhood lymphoma patients show the best long-term survival rates among thoracic tumors; therefore, they have a higher chance of developing chronic cardiovascular complications. Please summarize the epidemiologic data of these tumors in the Introduction if you choose to focus on these tumor types.

Thank you for your insightful comment. We agree with your observation and have addressed it by adding a summary of the epidemiologic data for lung cancer, breast cancer, and childhood lymphoma in the Introduction. This addition, included in lines 44–58, emphasizes the survival rates and long-term outcomes of these tumor types, providing a clearer rationale for our focus on their associated chronic cardiovascular complications.

  1. What was the basis for selecting preclinical studies using animal models and pharmacologic interventions against cardiotoxicity? Some highlighted studies can not comprehensively review the broad literature on cardiotoxic mechanisms and potential interventional opportunities. 

Thank you for your comment. The selection of preclinical studies using animal models and pharmacologic interventions was based on their relevance to the specific cardiotoxic mechanisms and interventional strategies discussed in our study. We prioritized studies that provided robust mechanistic insights, demonstrated translational potential, or focused on clinically relevant cardiotoxic effects induced by cancer therapies. While we aimed to highlight key studies to illustrate critical findings and trends, we acknowledge that the selected works cannot encompass the entire breadth of the literature. Our intention was not to provide an exhaustive review but rather to focus on studies that align with the scope and objectives of our manuscript.

  1. The tables should not contain whole sentences because they are the repetition of the main text. They should be expanded with further important studies. Please provide the references in a separate column within the tables. It would be helpful to give the species/type of animal, sex, age, follow-up time, therapeutic intervention (with regimen, if applicable), and main molecular mechanisms with keywords in the tables.

Thank you for your valuable feedback. In response to your suggestion, we have streamlined the tables by removing full sentences that repeated the main text. Additionally, we have created a separate column within the tables to include bibliographic references. The number of references has been significantly increased, from 138 to 217, to incorporate further important studies.

  1. Figures summarizing the molecular mechanisms and interactions of the different oncologic treatments are missing.

Thank you for your insightful comment. In response to your suggestion, we have included two new figures (Figures 3 and 4) to address the missing illustrations. Figure 3 depicts the progression from radiation-induced epithelial/endothelial damage to fibroatrophy, while Figure 4 illustrates the development of ICI-associated myocarditis. We believe these additions will provide a clearer understanding of the molecular mechanisms and interactions of the oncologic treatments discussed in the manuscript. We greatly appreciate your valuable input.

  1. Beyond age, sex, and gender, comorbidities and prescribed drugs can modify the (side) effects of the oncologic treatments. Please mention these factors also.

Thank you for your thoughtful comment. We agree that factors such as comorbidities and prescribed drugs can significantly influence the (side) effects of oncologic treatments. These aspects have now been explicitly addressed in the manuscript, specifically in lines 45–48, to provide a more comprehensive discussion.

____________________________________

Reviewer 2 Report

Comments and Suggestions for Authors

In this narrative review the authors summarized the published
experimental cell culture and animal models of cardiotoxicities in
chemotherapy, immunotherapy, and radiotherapy. The success rate
of novel anticancer approaches individual or in combination,
are not free from cardiotoxic effects. The conclusion of the
authors is that preclinical models are essential in advancing our
knowledge of cardiotoxicity.
Concerns
1. This review is focusing in experimental testing systems of
cardiotoxicity. There are no comparisons between the experimental
and clinical evidence to confierm the translational value of these
approaches. The title of the review is misleading as there are
no connections between the preclinical and clinical research.
2. The conclusion of the authors is too generic and ungrounded.
The described experimental cell culture of animal models have
not been compared with clinical data. It is not know in what
degree these preclinical models delivers the human heart condition
after chemotherapy, immunotherapy, or radiotherapy
3. Only one of the studies presented [ref#122] examined the critical
effects of the long-term consequences of a treatment exposure, in
particular radiation. This critical factor is underestimated in
preclinical studies. The focus is mostly in acute effects of
treatment.
4. There is no section 4 in the review. The transition from section
3.3 Immunotherapy to 4.1 Examples of in vitro and in vivo models...
after Table 5 has not the title of section 4.
5. The authors should add clinical reports that bridge the
presented pre-clinical studies with what happens to human patients.

Author Response

Reviewer 2

Comments and Suggestions for Authors

In this narrative review the authors summarized the published experimental cell culture and animal models of cardiotoxicities in chemotherapy, immunotherapy, and radiotherapy. The success rate of novel anticancer approaches individual or in combination, are not free from cardiotoxic effects. The conclusion of the authors is that preclinical models are essential in advancing our knowledge of cardiotoxicity.

Answer: We thank the Reviewer for his time spent reading and evaluating our manuscript. We agree with all the points raised (see subsequent answers) and we believe that thanks to his comments our manuscript has significantly improved.

Concerns

  1. This review is focusing in experimental testing systems of cardiotoxicity. There are no comparisons between the experimental and clinical evidence to confierm the translational value of these approaches. The title of the review is misleading as there are no connections between the preclinical and clinical research.
  2. We acknowledge that the previous version of the manuscript did not adequately highlight the link between preclinical models and clinical outcomes. We have now expanded the discussion (see Paragraphs 4.2.1, 4.3.1 and 4.4.1 )by including specific examples of how preclinical models have contributed to understanding and managing therapy-induced cardiotoxicity in oncology.

As highlighted in the review, many clinical studies use healthy animal models that do not reflect the clinical conditions of oncology patients, given that the tumor itself can impact the cardiovascular system. This underscores the importance of preclinical models that better mimic the complex pathophysiology of the disease. 

  1. The conclusion of the authors is too generic and ungrounded. The described experimental cell culture of animal models have not been compared with clinical data. It is not know in what degree these preclinical models delivers the human heart condition after chemotherapy, immunotherapy, or radiotherapy
  2. We acknowledge the reviewer's concern regarding the need for a more grounded conclusion that directly addresses the limitations of current preclinical models in comparison to clinical data. In response, we have revised the conclusion to explicitly highlight the gaps between preclinical findings and clinical outcomes, emphasizing the lack of robust comparative data that validate these models against real-world patient conditions. Specifically, we have added a paragraph that underscores the critical need for future research to bridge this translational gap by incorporating clinical data into preclinical studies. This includes the development of more sophisticated models, such as patient-derived iPSCs and organ-on-a-chip systems, which can better mimic human physiology and pathology. Additionally, we stress the importance of long-term follow-up studies in animal models to assess chronic cardiotoxic effects, as well as the integration of multi-omics approaches to provide a more comprehensive understanding of the mechanisms underlying therapy-induced cardiotoxicity. By addressing these limitations, we aim to build a more reliable and predictive platform that can ultimately improve the safety and efficacy of cancer therapies.

  1. Only one of the studies presented [ref#122] examined the critical effects of the long-term consequences of a treatment exposure, in particular radiation. This critical factor is underestimated in preclinical studies. The focus is mostly in acute effects of treatment.
  2. We agree with this comment that the long-term consequences of treatment, particularly radiation, are often underestimated in preclinical studies, which typically focus on the acute effects. However, it is challenging to conduct preclinical studies that investigate the long-term effects of ionizing radiation. Therefore, bridging preclinical findings with clinical studies becomes essential in understanding the full spectrum of treatment-related consequences.
  3. There is no section 4 in the review. The transition from section 3.3 Immunotherapy to 4.1 Examples of in vitro and in vivo models after Table 5 has not the title of section 4.
  4. We have added an introduction to Section 4. Thank you for this point.
  5. The authors should add clinical reports that bridge the presented pre-clinical studies with what happens to human patients.
  6. We have incorporated paragraphs 4.1.1, 4.2.1, and 4.3.1 to summarize the clinical reports that integrate findings from preclinical models.

Reviewer 3 Report

Comments and Suggestions for Authors

This review provides useful information for understanding the role of preclinical models in studying the cardiotoxicity of cancer drugs or radiation. The writing is good. Just one suggestion: zebrafish is now a quite popular model in cardiotoxicity study. So it would be better to add zebrafish in the Preclinical animal model section.

Author Response

Reviewer 3

This review provides useful information for understanding the role of preclinical models in studying the cardiotoxicity of cancer drugs or radiation. The writing is good. Just one suggestion: zebrafish is now a quite popular model in cardiotoxicity study. So it would be better to add zebrafish in the Preclinical animal model section.

We thank the reviewer for their valuable suggestion. We agree that zebrafish is indeed a popular and versatile model for studying cardiotoxicity. Following your recommendation, we have added a dedicated paragraph in the "Preclinical Animal Models" section (lines 227–236) discussing the utility of zebrafish in cardiotoxicity research. Additionally, this model is revisited later in the manuscript (lines 559–568) to highlight its specific applications and advantages in this context.

Round 2

Reviewer 1 Report

Comments and Suggestions for Authors

The authors have improved their MS; however, several issues should be addressed and modified according to the requests of the first review:

  1. I can not see the modifications in the revised MS because they are not highlighted or colored.  
  2. Title: please remove letter A from "ATranslational".
  3. References to the epidemiology data are missing (lines 44-58). Please add them to the text and reference list.
  4. A statement that clarifies the basis for selecting preclinical studies using animal models and pharmacologic interventions against cardiotoxicity is missing at the end of the introduction. 
  5. The authors did not respond appropriately to my original request 4: Tables contain whole sentences.  Table 2 mixes different studies together and classifies them according to the used species. Tables 3-5 did not provide information on species/type of animal, sex, age, follow-up time, or therapeutic intervention (with regimen, if applicable). This information is necessary for the readers to see sex-, dose-, age-, etc. dependent cardiotoxic effects. Please elaborate on Tables 2-5 according to the original request.

Author Response

The authors have improved their MS; however, several issues should be addressed and modified according to the requests of the first review:

  1. I can not see the modifications in the revised MS because they are not highlighted or colored.  

The highlighted version was submitted under “Supplementary Materials” section, I apologize for the misunderstanding.

  1. Title: please remove letter A from "ATranslational".

Thank you for your suggestion. The error has been corrected by removing the letter "A" from "ATranslational."

  1. References to the epidemiology data are missing (lines 44-58). Please add them to the text and reference

Thank you for your suggestion. The references to the epidemiological data have been added and are now numbered 1 to 12 in both the text and the reference list.

  1. A statement that clarifies the basis for selecting preclinical studies using animal models and pharmacologic interventions against cardiotoxicity is missing at the end of the introduction. 

Thank you for your comment. A clarification regarding the basis for selecting preclinical studies using animal models and pharmacologic interventions against cardiotoxicity has been added on lines 67-74.

  1. The authors did not respond appropriately to my original request 4: Tables contain whole sentences.  Table 2 mixes different studies together and classifies them according to the used species. Tables 3-5 did not provide information on species/type of animal, sex, age, follow-up time, or therapeutic intervention (with regimen, if applicable). This information is necessary for the readers to see sex-, dose-, age-, etc. dependent cardiotoxic effects. Please elaborate on Tables 2-5 according to the original request.

Thank you for your suggestion. The references have been checked and corrected, with particular attention given to Table 2, where some mixing of studies had occurred. The purpose of our tables remains to summarize the key points discussed in the text, rather than to include the detailed technical aspects of each study. It would be quite challenging to incorporate all the requested information, such as species, sex, age, follow-up time, and therapeutic intervention, into the tables while maintaining clarity and conciseness.